# Molecular Dynamics Study of the Curvature-Driven Interactions between Carbon-Based Nanoparticles and Amino Acids

**DOI:** 10.3390/molecules28020482

**Published:** 2023-01-04

**Authors:** Wanying Huang, Zhenyu Wang, Junyan Luo

**Affiliations:** 1T-Life Research Center, State Key Laboratory of Surface Physics, Department of Physics, Fudan University, Shanghai 200433, China; 2Zhejiang Lab, Nanhu-Kechuang Avenue, Yuhang District, Hangzhou 310000, China; 3Department of Physics, Zhejiang University of Science and Technology, Hangzhou 310000, China

**Keywords:** CBNs, amino acids, molecular dynamics

## Abstract

We researched the interaction between six representative carbon-based nanoparticles (CBNs) and 20 standard amino acids through all-atom molecular dynamics simulations. The six carbon-based nanoparticles are fullerene(C60), CNT55L3, CNT1010L3, CNT1515L3, CNT2020L3, and two-dimensional graphene (graphene33). Their curvatures decrease sequentially, and all of the CNTs are single-walled carbon nanotubes. We observed that as the curvature of CBNs decreases, the adsorption effect of the 20 amino acids with them has an increasing trend. In addition, we also used multi-dimensional clustering to analyze the adsorption effects of 20 amino acids on six carbon-based nanoparticles. We observed that the π–π interaction still plays an extremely important role in the adsorption of amino acids on carbon-based nanoparticles. Individual long-chain amino acids and “Benzene-like” Pro also have a strong adsorption effect on carbon-based nanoparticles.

## 1. Introduction

The interaction between nanoparticles and proteins has attracted widespread attention [1,2]. Molecular dynamics is very important in the study of nanobiology, especially the interaction between nanoparticles and proteins [3,4,5]. Experimental and theoretical simulation studies between nanoparticles and proteins [6] have been in continuous progress [7], which is of great significance to the research of biopharmaceuticals, especially targeted drugs [8]. Various gold nanoparticles modified with different peptide chain groups have become important research objects for cancer-targeted drugs [9]. In addition, recent interactions between two-dimensional nanomaterials and individual nucleotides [10] indicate that this two-dimensional screening structure of in-plane graphene and hexagonal boron nitride arrays may inspire the development of laboratory-on-a-chip sequencing in the future. Carbon-based nanoparticles, as the most basic nanoparticles, have been the wide concern of molecular dynamics research between them and proteins [11,12,13,14,15]. At present, many researchers have used molecular dynamics to study the interaction between carbon-based nanoparticles and proteins [8,16,17,18,19,20].

The interaction between amino acids and carbon-based nanoparticles or modified carbon-based nanoparticles has attracted wide attention from researchers in experimental, quantitative calculation, and molecular dynamics [13,15,21,22,23,24,25]. Experimentally, the interaction between carbon-based nanoparticles and amino acids, as well as the unique electronic and optical properties of single-walled carbon nanotubes [23,26], coupled with their size and mechanical strength, make single-walled carbon nanotubes the key to the sensing platform [12] to the development of the next generation of biological transmission. Experimental studies have shown that carbon-based nanomaterials can promote neuronal electrical signals [19]. In addition, researchers have developed a variety of electrochemical biosensors based on carbon nanotubes [20]. Moreover, in terms of drug delivery, carbon-based nanoparticles, especially those modified by peptides, have always received extensive attention from biomedical workers [27]. For quantum calculations, the study of amino acids or amino acids has occurred in a vacuum computing environment. The interaction between aromatics and carbon nanotubes [7,13,14], and this part of the research content, is also an important reference for the study of the interaction of 20 amino acids and carbon-based nanoparticles in aqueous solutions. For all-atom molecular dynamics, on the one hand, some researchers have studied the interaction between some amino acids and a single carbon nanotube in one system [17].

However, there is still a lack of molecular dynamics studies on the curvature-driven relationship between CBNs and amino acids. Through our research, a detailed study of the interaction between a single amino acid and six carbon-based nanoparticles at the full-atom level was conducted. We tried to explore the relationship between 20 standard amino acids and six kinds of carbon-based nanoparticles from two views. From the view of carbon-based nanoparticles and under the premise of excluding the volume limitation, will nanoparticles with different curvatures affect the adsorption effect between them and amino acids? From another view of 20 amino acids, in which different groups of amino acids are compared with six carbon-based nanoparticles, will the adsorption effect of a representative carbon-based nanoparticle be affected by its type? In addition, cluster analyses in machine learning can well characterize the adsorption strength between six representative carbon-based nanoparticles (CBNs) and 20 amino acids. The main interactions that carbon-based nanoparticles can have with amino acids are hydrophobic interactions and pi–pi interactions. Zuo’s article [8] elaborated: “the CNT binding on proteins is believed to be driven by various weak interactions such as π–π stacking, hydrophobic and electrostatic interactions.” Therefore, the driving force between carbon-based nanoparticles and amino acids is not the present work’s research focus.

The comprehensive all-atom simulation study between carbon-based nanoparticles and 20 representative amino acids is very important. (1) This work is an important preliminary work for multi-scale research between carbon-based nanoparticles and ultra-large proteins; (2) This research is a paradigm for the study of interactions between other nanomaterials and 20 amino acids, and it is also the basis for future research on the relationship between complex nanomaterials and their interactions; (3) This work is also a study on the binding of nanoparticles and amino-acid-related small molecules or proteins. It is also important supporting material for predictions. Next, the interactions between six representative carbon-based nanoparticles (CBNs) and 20 amino acids from the perspective of all-atom molecular dynamics simulations is explored.

## 2. Results

**Simulation system.** In our research, 120 molecular systems (6CBNs×20Aminoacids×3) were constructed, and the interaction between six carbon-based nanoparticles and 20 amino acids (Appendix A) in an aqueous solution (Appendix A) was studied. In each research, the system conducted three repeated simulations, with a total of more than 360 trajectories. Each trajectory was 200 ns, and we tried to objectively statistically research them from the all-atom scale. Among them, the six types of carbon-based nanoparticles were fullerene (C60), single-walled carbon nanotube (CNT55L3, CNT1010L3, CNT1515L3, CNT2020L3), and graphene (graphene33). In our research, C60 was the carbon-based nanoparticle with the largest curvature (1/r), and graphene33, which had the smallest curvature, was close to the planar state. In the following, we will first introduce the trend in adsorption between the CBNs and amino acids. We also discuss the effect of the CBNs’ curvature change on adsorption, and the effect of change in the amino acid side chain on adsorption.

### 2.1. CBNs Adsorption Trend Analysis

Our study found an interesting trend: that is, the smaller the curvature of the carbon-based nanoparticles, the better the adsorption of the amino acids. Figure 1A shows the six representative carbon-based nanoparticles. The curvatures of the CBNs we studied were different, as shown in Figure 1B. Their curvatures decreased continuously, from a curvature of C60 at 2.99401 ${\mathrm{nm}}^{-1}$ to a curvature that approached zero for graphene. As shown in Appendix A, the curvature span of our chosen carbon nanotubes was just between C60 and graphene.

The average adsorption strength can be seen in that, as the curvature of the CBNs in Figure 1B decreases, their respective overall adsorption rates to the amino acids tend to increase, as shown in Figure 1C. It shows the average adsorption strength of 20 amino acids for six kinds of CBNs. We can calculate the first-peak minimum-distance distribution of the CBNs obtained from Appendix A on the 20 amino acids; that is, the amino acids corresponding to the same CBNs are added and averaged. As a result, we are pleased to observe from Figure 1C that, as the curvature of the CBNs becomes smaller, the adsorption strength of the carbon-based nanoparticles of 20 amino acids had an obvious upward trend. This also shows that the average minimum-distance-distribution first peak can well describe the influence of CBNs’ curvature changes on the adsorption strength of 20 amino acids. In other words, the surface curvature of carbon-based nanoparticles has an obvious effect on the adsorption between them and amino acids.

### 2.2. Kinetics and Statistical Analysis

This paper mainly studies the adsorption trend between six representative carbon-based nanoparticles and 20 standard amino acids, as shown in Figure 2A. Our research found that as the curvature decreases, the adsorption rate shows an increasing trend, as shown in Figure 2B. Moreover, our research also makes suitable use of multi-dimensional clustering analysis to describe the overall change in its adsorption rate when 20 amino acids change with the decreasing curvature of carbon-based nanoparticles.

As shown in Figure 2C, the purple line is the short-chain amino acid 01-Gly, the yellow line is the hydrophobic amino acid 07-Met with the longest side chain, and the green line is the aromatic amino acid 15-Trp with the longest side chain: (1) **non-polar hydrophobic short-chain amino acid 01-Gly.** The contact between Gly and various CBNs is extremely unstable, as shown in Figure 2B(a). This unstable adsorption is also reflected in Mindist-PDF, and their first-peak values are all less than 0.15, as shown in Figure 2A: (2) **non-polar hydrophobic long-chain amino acids 07-Met.** The adsorption effect of 07-Met is moderate, as shown in Figure 2B(b). The adsorption effect between 07-Met and CBNs increases as the curvature decreases, as shown in Figure 2C: (3) **aromatic amino acid 15-Trp.** 15-Trp is an amino acid with the best adsorption effect on CBNs among aromatics, and it has a “bicyclic” side chain structure. The adsorption effect between 15-Trp and CBNs is relatively good.

We can observe that the adsorption strength of the three amino acids is as follows: 15-Trp > 07-Met > 01-Gly, as shown on Figure 2C. In other words, the adsorption strength of aromatic amino acids is much greater than that of amino acids with the longest side chain.

### 2.3. Multi-Dimensional Cluster Analysis

Our research separately counted the situation when n_cluster = 2,3,4,5,6,7. For example, during the process of changing from red to black in Figure 3a, the clustering value decreased successively. When n_cluster = 2, the amino acids in the upper region included the following: special hydrophobic amino acid 03-Pro; long-chain hydrophobic amino acid 05-Leu; 06-Ile, 07-Met; long-chain uncharged amino acid 11-Asn; 12-Glu; an all-aromatic group of amino acids 13-Phe, 14-Tyr, and 15-Trp; and partially positively charged amino acids 17-Arg and 18-His. When n_cluster = 3, as shown in Figure 3b, long-chain hydrophobic amino acids (05-Leu, 06-Ile, 07-Met) and long-chain uncharged amino acids (11-Asn, 12-Glu) fell out of the first echelon. When n_cluster = 4, the first echelon remained unchanged, and the lower zone began to split. When n_cluster = 4, as shown in Figure 3c, the median region began to change. The first echelon of the upper region was the red special hydrophobic amino acid 03-Pro, the long-chain aromatic amino acid 15-Trp, and the positively charged amino acid 17-Arg. The second echelon of the upper region was the orange aromatic amino acid 13-Phe and 14-Tyr. Next, when n_cluster = 5, 6, 7, as shown in Figure 3d–f, and regardless of which region, the amino acid arrangement changed and the first and second echelon of the upper region remained unchanged. The first echelon of the upper region was the red special hydrophobic amino acid 03-Pro, the long-chain aromatic amino acid 15-Trp, the positively charged amino acid 17-Arg, and the second echelon of the upper region is the orange aromatic amino acid 13-Phe and 14-Tyr.

In other words, the special hydrophobic amino acid 03-Pro, the long-chain aromatic amino acid 15-Trp, and the positively charged amino acid 17-Arg are far ahead of the adsorption capacity of these three amino acids on carbon-based nanoparticles. Regarding the remaining two aromatic amino acids, the adsorption effect of 13-Phe and 14-Tyr is better than that of other amino acids.

The binding strength of amino acids to carbon-based nanoparticles is limited to graphene. Therefore, as shown in Figure 4, we can observe the situation of RMSF and RMSD of three representative amino acids at the binding limit. For RMSF, the better hydrophobic effect of the side chain, the stronger the fluctuation of RMSF. It is worth noting that the RMSF value of the middle atom of 15-Trp is higher than that of the other two amino acids. That is to say, aromatic amino acids are absorbed mainly by their own strength, regardless of the modification of both ends. While 01-Gly is obviously driven by the modified groups at both ends, for RMSD, the fluctuation range of amino acid RMSD value is different. The RMSD value of 01-Gly fluctuates around 0.06 nm, that of 07-Met fluctuates around 0.13 nm, and that of 15-Trp fluctuates around 0.175 nm.

The trend in RMSD of aromatic amino acids changes with time, as shown in Appendix A. When 15-Trp interacts with different carbon-based nanoparticles, the numerical range of RMSD is obviously different. It is worth noting that the better the adsorption effect is, the more periodic the RMSD value of 15-Trp will be.

## 3. Discussion

We studied the interactions between six representative carbon-based nanoparticles (CBNs) and 20 amino acids through all-atom molecular dynamics simulations. A total of 120 molecular systems (6CBNs×20Aminoacids) were constructed, the interactions between six carbon-based nanoparticles and 20 amino acids in aqueous solutions were studied, and three repeated simulation studies for each research system were performed, totaling more than 360 simulation research trajectories. The simulation time for each trajectory was 200 ns, as we tried to objectively statistically research them from the all-atom level. It was determined that the interaction between them was driven by curvature. The hydrophobic surface of the CBNs was determined, and the surface curvature was the key to the adsorption strength between it and amino acids.

## 4. Materials and Methods

**(1)** 
**Simulate system MD parameters**


A molecular system of 120 simulation systems (6CBNs×20Aminoacids) in an aqueous solution is shown in Figure 1A and Appendix A. We selected the most commonly used SPC model for water molecules and the Amber03 force field for amino acids. Among them, the larger carbon-based nanoparticles were the CNT2020L3 system, and we added 21,245 water molecules. We used Gromacs software [28,29] throughout our molecular dynamics research. Our study used the frog-leap method simulation, and the simulation integration step was 2 fs. To ensure a sufficient adsorption time between the amino acids and CBNs, the duration of each simulation was 200 ns. We ensured that our all-atom simulation time was sufficient to extract sufficient conformational information to prepare for the subsequent multi-scale coarse-grained force field. In the initial stage, the distance between the amino acids and CBNs was about 1.5 nm. Our simulation systems were all NVT systems [30]. The simulated temperature of the system was 330K, and we used the Berendsen temperature-adjustment method. We used periodic boundary conditions to make our carbon nanotubes a periodic infinite length and an infinite graphene plane [29]. Additionally, the smooth cutoff had a distance value of 1nm. Notably, in order to eliminate the influence of the CBNs’ volume on the adsorption effect, all simulation systems had the same surface contact volume ratio, as shown in Appendix A. In Appendix A, more molecular simulation schematic details are shown.

**(2)** 
**Analysis method**


In our study, there were three research parameters that were the key to characterizing the adsorption strength between CBNs and amino acids. They are as follows: (1) Mindist: the minimum contact distance, and the mindist changed with time for each group of amino acids; (2) Mindist-PDF: the study counted the closest distances within 3 nm, calculated the distribution of the minimum distances, and normalized them; (3) Adsorption index: the first-peak of Mindist-PDF was selected as an important indicator to describe the adsorption strength between the CBNs and amino acids. In addition, we selected multi-dimensional clustering (sklearn.cluster.KMeans) to study the adsorption strength between CBNs and 20 representative amino acids.

The change in the minimum contact distance with time is a very important indicator. In many studies of proteins and nanoparticles, this indicator is used to describe the change in distance from the surface of the nanoparticle with time. The reason for this is that different carbon-based nanoparticles have different radii and choosing the mindist indicator can better reflect the adsorption between amino acids and carbon-based nanoparticles.

## 5. Conclusions

We observed that as the curvature of carbon-based nanoparticles decreased, the adsorption effect of the 20 amino acids with them continuously improved. In addition, we used multi-dimensional clustering to analyze the adsorption effects of 20 amino acids on six carbon-based nanoparticles. We observed that the π–π interaction plays an extremely important role in the adsorption of amino acids on carbon-based nanoparticles. Individual long-chain amino acids also have a strong adsorption effect on carbon-based nanoparticles. Specifically, the special hydrophobic amino acid 03-Pro, the long-chain aromatic amino acid 15-Trp, and the positively charged amino acid 17-Arg are three amino acids that are far ahead in terms of the adsorption capacity of carbon-based nanoparticles. Regarding the remaining two aromatic amino acids, the adsorption effect of 13-Phe and 14-Tyr is better than that of other amino acids.

In addition, the advent of AlphaFold/Fold2 [31] and ESM-Fold [32] has brought protein folding or prediction into the fast lane of research, which means that it is very important to include artificial intelligence in the study of the dynamic behavior mechanism of a large number of proteins. Additionally, some forms of artificial intelligence reinforcement learning, such as Bayesian optimization [33], requires some prior knowledge of prediction. The binding between proteins and amino acids is very dependent on the binding pocket of the protein. Most of the proteins are mainly of a hydrophobic core. Therefore, this study characterized the binding of carbon-based nanoparticles of different sizes to 20 representative amino acids.

Our research results can be applied to the following three aspects in the future: (1) The regular adsorption effect between CBNs and amino acids can bring certain inspiration to the new generation of nano-biocomputers; (2) This research work as a full-atom simulation of carbon-based nanoparticles and 20 kinds of amino acids has laid a solid simulation data foundation for further multi-scale coarse-grained force-field extraction; (3) Researchers can select different binding clustering results according to the binding pocket size of the research protein as a priori judgment for some related reinforcement learning research.

In previous studies, there was no such comprehensive study on the large-scale molecular dynamics study between carbon nanoparticles of different sizes and 20 kinds of amino acids. We studied 6*20*3 groups of studies. Moreover, the characterization index mindist in this study can be well transplanted to the study of large proteins and carbon-based nanoparticles. The innovation of this study is that, based on the molecular dynamics simulation results, it is very interesting to observe that the adsorption strength between carbon-based nanoparticles and amino acids is driven by the curvature of carbon-based nanoparticles.

## Figures and Tables

**Figure 1 molecules-28-00482-f001:**
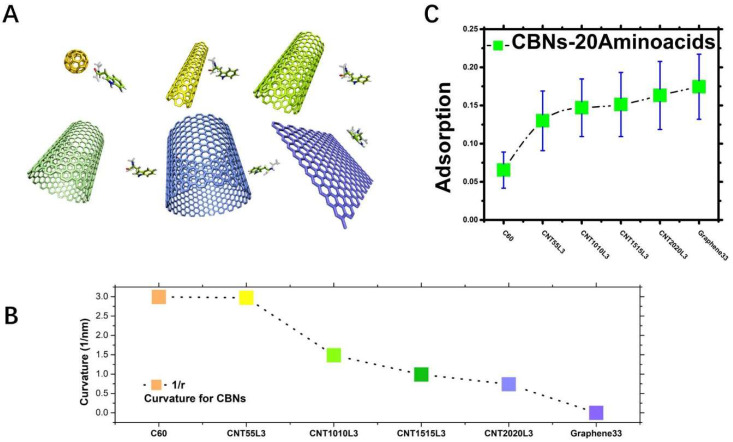
(**A**). Six representative carbon-based nanoparticles; (**B**). The curvatures of six representative CBNs; (**C**). The average adsorption strength of six CBNs for 20 amino acids. The green square is the average adsorption strength of each carbon-based nanoparticle for 20 amino acids, and the blue bar line is the corresponding error.

**Figure 2 molecules-28-00482-f002:**
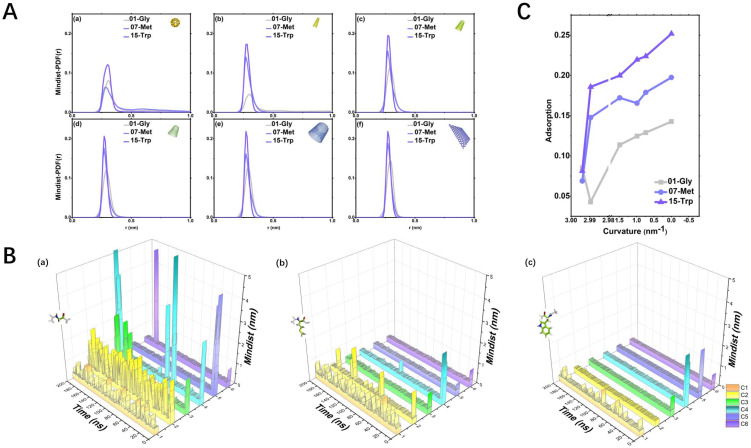
(**A**). Mindist-PDF(r): the distribution of minimum distances between three representative amino acids and CBNs. The grey line is 01-Gly, the blue line is 07-Met, and the purple line is 15-Trp; (**B**). Mindist: Variation of the minimum distances between three representative amino acids (a–c) and six CBNs with time. C1 orange line is C60; C2 yellow line is CNT55L3; C3 green line is CNT1010L3; C4 bright blue line is CNT1515L3; C5 blue-purple line is CNT2020L3; C6 purple line is graphene33; (**C**). Adsorption: The adsorption trend in three representative amino acids, the grey dotted line is 01-Gly, the blue dotted line is 07-Met, and the purple dotted line is 15-Trp.

**Figure 3 molecules-28-00482-f003:**
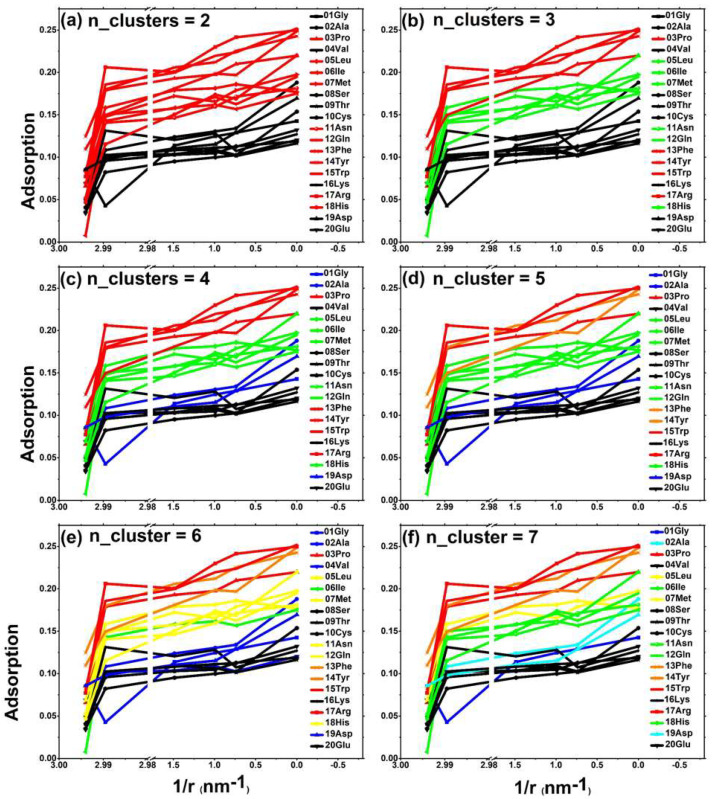
Multi-dimensional clustering analysis (the horizontal axis of this group of graphs is the curvature corresponding to the six types of carbon-based nanoparticles), clustering the peak distribution peaks of the 20 amino acids to the nearest distance of the carbon-based nanoparticles: (**a**) n_cluster = 2; (**b**) n_cluster = 3; (**c**) n_cluster = 4; (**d**) n_cluster = 5; (**e**) n_cluster = 6; (**f**) n_cluster = 7. The colors of the lines with cluster scores from high to low are a red dotted line, orange dotted line, yellow dotted line, green dotted line, bright blue dotted line, dark blue dotted line, and black dotted line.

**Figure 4 molecules-28-00482-f004:**
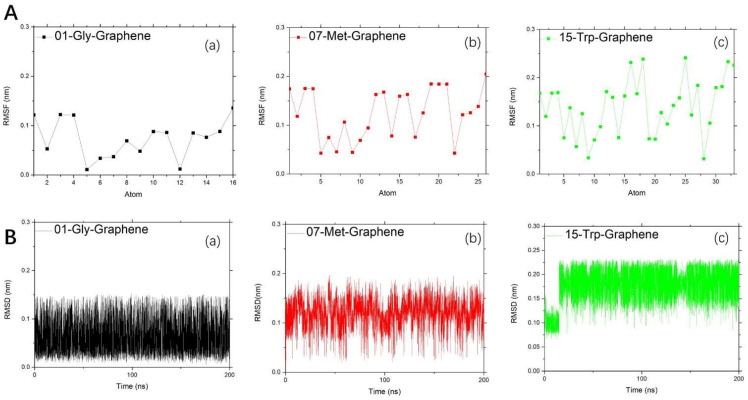
Study on the stability between representative amino acids and graphene: (**A**). Research on RMSF: (a) black dotted line for 01-Gly-Graphene; (b) red dotted line for 07-Met-Graphene; and (c) green dotted line for 15-Trp-Graphene; (**B**). Research on RMSD change within time: (a) black line for 01-Gly-Graphene; (b) red line for 07-Met-Graphene; and (c) green line for 15-Trp-Graphene.

## Data Availability

Data is contained within the article or Appendix A. The data presented in this study are available in insert article or Appendix A here.

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
