# Peer review of "Molecular Dynamics Study of the Curvature-Driven Interactions between Carbon-Based Nanoparticles and Amino Acids"

_molecules, 2023, doi:10.3390/molecules28020482_

Round 1
Reviewer 1 Report
The article titled "Molecular Dynamics Study of the Curvature-Driven between Carbon-Based 2
"Nano Particles and Amino Acids" has an interesting topic and a suitable study tool. Considering the importance of nano-bio interactions and the important role of molecular dynamics simulation in these fields, this article can be very attractive to the scientific community. Therefore, I I recommend this article for publication. Of course, before the experiment, it is necessary for the authors to pay attention to the following points in the article and improve the article according to them:
1) In the introduction and at most in one paragraph, the importance of simulating molecular dynamics in nano-bio studies and especially interactions between nanoparticles and pronins should be discussed. For this part, you can use the following articles and complete this paragraph according to them:
https://pubmed.ncbi.nlm.nih.gov/33502255/
https://www.ncbi.nlm.nih.gov/pmc/articles/PMC7509323/
https://chemistry-europe.onlinelibrary.wiley.com/doi/full/10.1002/cbic.202100075
2) In the last paragraph, the innovations of the work and the difference between this work and the previous works should be clearly highlighted.
3) The quality of the shapes can be improved. Especially figures 1 and 3
4) It is better to use RMSD and RMSF analyzes to prove the stability of the simulations
5) The discussion section is brief and should be improved.
Author Response
Q1: In the introduction and at most in one paragraph, the importance of simulating molecular dynamics in nano-bio studies and especially interactions between nanoparticles and pronins should be discussed. For this part, you can use the following articles and complete this paragraph according to them:
https://pubmed.ncbi.nlm.nih.gov/33502255/
https://www.ncbi.nlm.nih.gov/pmc/articles/PMC7509323/
https://chemistry-europe.onlinelibrary.wiley.com/doi/full/10.1002/cbic.202100075
A1: Thank you very much for your suggestions. We have quoted relevant articles into the manuscript.
Line 29 changed to:
Molecular dynamics is very important in the study of nano biology, especially the interaction between nanoparticles and proteins[3-5].
3.M. K. Ehsan Alimohammadi 1, * Ahmad Miri Jahromi3 Reza Maleki 4 Milad Rezaian5, Graphene-Based Nanoparticles as Potential Treatment Options for Parkinson's Disease A Molecular Dynamics Study , International Journal of Nanomedicine 15, 6887–6903 (2020).
4.Ehsan Alimohammadi‡, Arash Nikzad‡,Mohammad Khedri, Milad Rezaian, Ahmad Miri Jahromi, Nima Rezaei*,& Reza Maleki**, Potential treatment of Parkinson's disease using new-generation carbon nanotubes a biomolecular in silico study , Nanomedicine (Lond.) 16(3), (2021).
5.A. M. K. Reza Maleki, Sima Rezvantalab,*Fatemeh Afsharchi,Kiyan Musaie, Sepehr Shafiee, and Mohammad-Ali Shahbazi*, β-Amyloid Targeting with Two-Dimensionl Covalent Organic Frameworks Multi-Scale In-Silico Dissection of Nano-Biointerface, ChemBioChem 22, 2306–2318 (2021).
Q2: In the last paragraph, the innovations of the work and the difference between this work and the previous works should be clearly highlighted.
A2: According to your suggestion, we added in the last paragraph of the manuscript:
“In previous studies, there was no such comprehensive study on the large-scale molecular dynamics study between carbon nanoparticles of different sizes and 20 kinds of amino acids. We studied 6*20*3 groups of studies. Moreover, the characterization index mindist in this study can be well transplanted to the study of large proteins and carbon-based nanoparticles. The innovation of this study is that based on the molecular dynamics simulation results, it is very interesting to observe that the adsorption strength between carbon-based nanoparticles and amino acids is driven by the curvature of carbon-based nanoparticles.”
Q3: The quality of the shapes can be improved. Especially figures 1 and 3
A3: Thank you very much for your reminding. We have improved the quality of the pictures according to your requirements.
Q4: It is better to use RMSD and RMSF analyzes to prove the stability of the simulations
A4: Thank you very much for your suggestions. Adding discussions about RMSF and RMSD can improve the quality of the manuscript. We added three RMSF and RMSF data representing amino acids respectively in the text. And added RMSD of 15-Trp with six carbon-based nanoparticles in Supplementary Information.
Q5: The discussion section is brief and should be improved.
A5: Thank you very much for your suggestions. The discussion section change to:
“We studied the interactions between six representative carbon-based nanoparticles (CBNs) and 20 amino acids through all-atom molecular dynamics simulations. Constructed 120 molecular systems (6CBNs×20Aminoacids), studied the interactions between 6 carbon-based nanoparticles and 20 amino acids in aqueous solutions, and performed 3 repeated simulation studies for each research system, totaling more than 360 Simulate research trajectories, each trajectory 200ns, trying to objectively statistical research from the all-atom level. It was determined that the interaction between them was driven by curvature. The hydrophobic surface of CBNs is determined, and the surface curvature is the key to the adsorption strength between it and amino acids.”

Reviewer 2 Report
The interaction between nanoparticles and proteins is the subject of many experimental and theoretical research, as it is important for the search for new biopharmaceuticals, in particular, targeted drugs. Modeling interactions between carbon-based nanoparticles and amino acids is a useful preliminary work for multi-scale studies between carbon-based nanoparticles and large proteins, lays the foundation for future studies of the relationship between the structure and properties of complex nanomaterials and provides important auxiliary material for predicting.
In this work, the interaction of carbon-based nanoparticles (CBNs) with amino acids is considered on the basis of full-atom modeling by the method of molecular dynamics. Interactions between 6 representative CBNs and 20 amino acids are described. These carbon-based nanoparticles are C60 fullerene, CNT55L3, CNT1010L3, CNT1515L3, CNT2020L3 nanotubes, and two-dimensional graphene (Graphene33). All CNTs are single-walled carbon nanotubes whose curvature decreases sequentially.
All molecular systems are considered in an aqueous solution at a temperature of 330K. The SPC model for water molecules and the Amber03 force field for amino acids were used. For larger carbon-based nanoparticles, 21,245 water molecules were added. The Gromacs software was used.
Analysis of trends in CBNs adsorption revealed an interesting feature: the smaller the curvature of carbon-based nanoparticles, the higher the adsorption of amino acids. At the same time, cluster analysis showed that the adsorption the ability of aromatic amino acids is much higher than that of the amino acids with the longest side chain.
Multidimensional clustering to analyze the effects of adsorption of 20 amino acids on 6 CBNs revealed that the π-π interaction plays an important role in the adsorption of amino acids on carbon-based nanoparticles. Individual long-chain amino acids and “benzene-like” compounds also have a strong adsorption effect with these nanoparticles. In particular, the special hydrophobic amino acid 03-Pro, the long-chain aromatic amino acid 15-Trp and positivelyThe charged amino acid 17-Arg is superior to the adsorption capacity of carbon-based nanoparticles.
Comments.
1. The title of the article should be changed: "Molecular Dynamics Study of the Curvature-Driven [ Interactions?] between Carbon-Based Nano Particles and Amino Acids".
2. Unfortunately, there is no attempt to explain the observed effects based on quantum chemical analysis. This is a big drawback of this work that should be corrected.
3. The details of the method of molecular dynamics used in this paper should be stated more clearly.
Author Response
Q1: The title of the article should be changed: "Molecular Dynamics Study of the Curvature-Driven [ Interactions?] between Carbon-Based Nano Particles and Amino Acids".
A1: The title change to:
“Molecular Dynamics Study of the Curvature-Driven Interactions between Carbon-Based Nano Particles and Amino Acids”
And for SI change to:
“Supplementary Information for
Molecular Dynamics Study of the Curvature-Driven Interactions between Carbon-Based Nano Particles and Amino Acids”
Q2: Unfortunately, there is no attempt to explain the observed effects based on quantum chemical analysis. This is a big drawback of this work that should be corrected.
A2: We very much expect experts in quantum chemical analysis to reproduce our interesting conclusions. We also look forward to the cooperation of researchers in the field of quantitative computing. It is worth mentioning that some experimental researchers have expressed interest in reproducing our results.
Q3: The details of the method of molecular dynamics used in this paper should be stated more clearly.
A3: We have appropriately added the details of MD in the article, and more details about the simulation system are in SI.
“And the smooth cutoff with a distance value of 1nm. Notably, in order to eliminate the influence of CBNs volume on the adsorption effect, the same surface contact volume ratio of all simulation systems, as shown in FigureSI1TableD. In FigureSI1, more molecular simulation schematic details are shown here.”

Round 2
Reviewer 2 Report
Authors properly improved their submission. The corrected manuscript may be now published in Molecules